# A Circulating Risk Score, Based on Combined Expression of Exo-miR-130a-3p and Fibrinopeptide A, as Predictive Biomarker of Relapse in Resectable Non-Small Cell Lung Cancer Patients

**DOI:** 10.3390/cancers14143412

**Published:** 2022-07-14

**Authors:** Silvia Marconi, Michela Croce, Giovanna Chiorino, Giovanni Rossi, Francesca Guana, Aldo Profumo, Paola Ostano, Angela Alama, Luca Longo, Giuseppa De Luca, Mariella Dono, Maria Giovanna Dal Bello, Marco Ponassi, Camillo Rosano, Paolo Romano, Zita Cavalieri, Massimiliano Grassi, Marco Tagliamento, Lodovica Zullo, Consuelo Venturi, Chiara Dellepiane, Luca Mastracci, Elisa Bennicelli, Paolo Pronzato, Carlo Genova, Simona Coco

**Affiliations:** 1Lung Cancer Unit, IRCCS Ospedale Policlinico San Martino, 16132 Genova, Italy; silvia.marconi@hsanmartino.it (S.M.); giovanni.rossi@hsanmartino.it (G.R.); angela.alama@hsanmartino.it (A.A.); luca.longo@hsanmartino.it (L.L.); mariagiovanna.dalbello@hsanmartino.it (M.G.D.B.); zita.cavalieri@hsanmartino.it (Z.C.); s4712787@studenti.unige.it (L.Z.); chiara.dellepiane@hsanmartino.it (C.D.); elisa.bennicelli@hsanmartino.it (E.B.); 2Biotherapies Unit, IRCCS Ospedale Policlinico San Martino, 16132 Genova, Italy; michela.croce@hsanmartino.it; 3Laboratory of Cancer Genomics, Fondazione Edo ed Elvo Tempia, 13900 Biella, Italy; francesca.guana@fondazionetempia.org (F.G.); paola.ostano@fondazionetempia.org (P.O.); 4Department of Medical, Surgical and Experimental Sciences, University of Sassari, 07100 Sassari, Italy; 5Proteomics and Mass Spectrometry Unit, IRCCS Ospedale Policlinico San Martino, 16132 Genova, Italy; aldo.profumo@hsanmartino.it (A.P.); marco.ponassi@hsanmartino.it (M.P.); camillo.rosano@hsanmartino.it (C.R.); paolo.romano@hsanmartino.it (P.R.); 6Molecular Diagnostic Unit, IRCCS Ospedale Policlinico San Martino, 16132 Genova, Italy; giuseppa.deluca@hsanmartino.it (G.D.L.); maria.dono@hsanmartino.it (M.D.); 7Medical Oncology Unit 1, IRCCS Ospedale Policlinico San Martino, 16132 Genova, Italy; massimiliano.grassi@hsanmartino.it; 8Department of Internal Medicine and Medical Specialties, University of Genova, 16132 Genova, Italy; marco.tagliamento@edu.unige.it (M.T.); carlo.genova@hsanmartino.it (C.G.); 9Pathology Unit, IRCCS Ospedale Policlinico San Martino, 16132 Genova, Italy; consuelobarbara.venturi@hsanmartino.it (C.V.); luca.mastracci@unige.it (L.M.); 10Department of Surgical Sciences and Integrated Diagnostic, University of Genova, 16132 Genova, Italy; 11Medical Oncology 2 Unit, IRCCS Ospedale Policlinico San Martino, 16132 Genova, Italy; paolo.pronzato@hsanmartino.it; 12Academic Medical Oncology Unit, IRCCS Ospedale Policlinico San Martino, 16132 Genova, Italy

**Keywords:** early-stage non-small cell lung cancer (NSCLC), liquid biopsy, exosome-miRNA, peptidome, fibrinopeptide A, prognostic signature, miR-130a-3p, risk score, disease recurrence

## Abstract

**Simple Summary:**

To date, the five-year survival rate of early stages of non-small cell lung cancer (NSCLC) is still disappointing and reliable prognostic factors are mandatory. Here, we performed in-depth high-throughput analyses of plasma circulating markers, including exosomal microRNAs and peptidome to identify a prognostic score. The miRnome profile selected the Exo-miR-130a-3p as the most overexpressed in relapsed patients. Peptidome analysis identified four progressively more degraded forms of fibrinopeptide A (FpA), which were depleted in relapse patients. Notably, a stepwise algorithm selected Exo-miR-130a-3p and the greatest FpA (2–16) to build a prognostic score, where high-risk patients had 18 months of median disease-free survival. Overexpression of miR-130a-3p cells led to a deregulation of pathways such as angiogenesis as well as the coagulation and metalloprotease, which might be linked to FpA reduction. The risk score integrating circulating markers may help clinicians predict early-stage NSCLC patients who are more likely to relapse after surgery.

**Abstract:**

To date, the 5-year overall survival rate of 60% for early-stage non-small cell lung cancer (NSCLC) is still unsatisfactory. Therefore, reliable prognostic factors are needed. Growing evidence shows that cancer progression may depend on an interconnection between cancer cells and the surrounding tumor microenvironment; hence, circulating molecules may represent promising markers of cancer recurrence. In order to identify a prognostic score, we performed in-depth high-throughput analyses of plasma circulating markers, including exosomal microRNAs (Exo-miR) and peptides, in 67 radically resected NSCLCs. The miRnome profile selected the Exo-miR-130a-3p as the most overexpressed in relapsed patients. Peptidome analysis identified four progressively more degraded forms of fibrinopeptide A (FpA), which were depleted in progressing patients. Notably, stepwise Cox regression analysis selected Exo-miR-130a-3p and the greatest FpA (2-16) to build a score predictive of recurrence, where high-risk patients had 18 months of median disease-free survival. Moreover, in vitro transfections showed that higher levels of miR-130a-3p lead to a deregulation of pathways involved in metastasis and angiogenesis, including the coagulation process and metalloprotease increase which might be linked to FpA reduction. In conclusion, by integrating circulating markers, the identified risk score may help clinicians predict early-stage NSCLC patients who are more likely to relapse after primary surgery.

## 1. Introduction

Lung cancer is a major cause of cancer mortality worldwide with an estimated 1.8 million deaths in 2020 [1]. Standard treatments for lung cancer include surgery, chemotherapy, radiotherapy, as well as molecularly targeted therapies and immunotherapy [2,3]. Surgery is the therapeutic option of choice for early-stage non-small cell lung cancer (NSCLC), although only less than 20% of NSCLC cases are diagnosed when the disease is confined to the primary site. Unfortunately, even in the case of localized, radically resected NSCLC, approximately one-third of patients experience disease recurrence; therefore, the global 5-year overall survival (OS) rate of 60% is still unsatisfactory [4]. Currently, the main prognostic factor in radically resected NSCLC is represented by the post-operative stage; indeed, adjuvant chemotherapy is currently proposed if lymph node involvement or a tumor greater than 4.0 cm are confirmed. However, the global benefit from adjuvant chemotherapy is limited, being a 5-year absolute survival benefit equal to 4% [5]. Notably, apart from the post-operative stage and the performance status, reliable prognostic factors able to predict the benefit from adjuvant treatments are still lacking [6]. Therefore, the identification of new biological biomarkers capable of predicting clinical outcome and improving the management of patients with resected lung cancer remains a hitherto unmet need. To date, growing evidence shows that cancer progression might depend on a dual interconnection between cancer cells and the surrounding tumor microenvironment (TME) [7,8]. Liquid biopsy, intended as a minimally invasive and accessible tool, is an alternative method to tumor biopsy to investigate the connection between cancer cells and TME in this framework [9]. Among all circulating bio-sources, extracellular vesicles, including exosomes, can be released from all cells, representing a source of unique information on TME [10]. In particular, it has been suggested that they would mediate the intercellular crosstalk in both physiological and pathological conditions through the transfer of specific genetic molecules. Exosomal microRNAs (Exo-miRs), a class of regulatory elements of post-transcriptional gene expression, are the most enriched and most frequently studied as biomarkers [11]. In addition to Exo-miRs, a low molecular weight protein fraction (LMW) can be found in biological fluids, and specific proteolytic degradation patterns have been correlated with patient outcomes in oncology. The origin of these circulating peptides identified in cancer patients is still a matter of scientific debate, although it has been suggested that they might derive from degradation of proteins involved in pathways activated by the primary tumor [12]. In the last decade, the analysis of the serum peptidome has become a topic of great interest in the study of some cancers including NSCLC [12,13]. Both Exo-miRs [14,15,16,17] and peptidome [18] have been proposed as promising prognostic biomarkers in NSCLC, although without reliable results, especially when considered individually. Indeed, increasing evidence supports the hypothesis that the combination of blood-based biomarkers can provide more accurate prognostic scores [19,20].

The aim of this study was to investigate the combined role of exo-miRs and peptidome, in association with clinical parameters, to predict disease recurrence of early-stage NSCLC patients surgically treated with curative intent. This composite prognostic signature might potentially allow the selection of patients at “high risk” of relapse to be proposed adjuvant treatments and who deserve post-surgical follow-up at shorter regular intervals. In addition, in vitro studies in NSCLC cell lines were performed to evaluate the effects of Exo-miRs.

## 2. Materials and Methods

### 2.1. Patient Selection and Biological Sample Collection

Biological samples (i.e., serum and plasma) from a cohort of 110 patients who had undergone surgery for a lung malignancy between 2009 and 2012 were obtained from the Biological Resource Center (CRB-HSM) of the IRCCS Ospedale Policlinico San Martino (Genova, Italy). The timespan was established in order to ensure that all the evaluated patients had an adequate follow-up for OS. All the biological samples were collected before surgical resection. Exclusion criteria were: (i) lung metastases from other malignancies; (ii) neoplasms with neuroendocrine differentiation (due to their different prognosis from adenocarcinomas and squamous-cell carcinoma); (iii) previous neoadjuvant therapy for NSCLC (due to its potential effect on the Exo-miR enrichment); (iv) death from a cause other than NSCLC; (v) unresectable IIIB and IV stage NSCLC (Tumor, Node, Metastasis (TNM) 7th International Union Against Cancer (UICC) Edition) [21]. The present study has been approved by the Local Ethics Committee (126/2019) and conducted in compliance with the provisions of the Declaration of Helsinki. For each patient included in the study, a written informed consent was obtained.

### 2.2. Microarray Analysis

Exo total RNA was purified from 0.5 mL of plasma using the ExoRNeasy Midi Kit (Qiagen, Hilden, Germany) and the miR presence was assessed by Qubit™ using the microRNA Assay Kit (Thermo Fisher Scientific, San Jose, CA, USA). The Exo-miR expression was profiled according to our optimized protocol [22]. Briefly, 8 µL of Exo-RNA with the spike-in controls were labeled and hybridized on SurePrint Human miR Microarrays 8 × 60 K (Agilent Technologies, Santa Clara, CA, USA; AMADID: 070156). We also added a synthetic 30-mer DNA poly-A oligonucleotide 3′ labeled pCp-Cy3 (50 amol) (TIB Molbiol SRL, Genova, Italy) to each hybridization mix to improve the Bright Corner signals for the automatic gridding. After 24 h of hybridization, the fluorescent signals of microarray were acquired using the G2565CA scanner (Agilent Technologies) and images were processed by Feature Extraction software v.9.5.3.1 (Agilent Technologies). Raw and processed data are available in the Gene Expression Omnibus (GEO; https://www.ncbi.nlm.nih.gov/geo/; accessed on 1 May 2022, ID: GSE198958). Analyses were performed using R software environment for statistical computing. Pre-processing, filtering and a differential expression analysis were carried out using the Limma package for microarray analysis, using the normexp method with an offset = 20 for background correction and the scale method for between array normalization. Replicated probes for the same miR were then averaged. Pre-filtering based on logIntensity >7 was performed before carrying out statistical analyses.

### 2.3. MiR Expression Assessment by Droplet Digital and Quantitative PCR

The absolute quantification of prognostic Exo-miRs was assessed by QX200 Droplet Digital PCR System (ddPCR) (Bio-Rad Laboratories, Hercules, CA, USA) as already described [23]. Briefly, 5.5 µL of Exo-RNA was reverse-transcribed (RT) using the miRCURY LNA RT Kit including the spike-in RNA control UniSp6 (Qiagen). Then, 4.5–9 µL of the diluted RT templates (1:50–5000) was amplified using the ddPCR™ EvaGreen Supermix (Bio-Rad Laboratories) and the specific miRCURY LNA miR PCR Assay (Appendix A). The data were analyzed using the QuantaSoft software (Bio-Rad). Each sample was run in duplicate with a negative template control (NTC) and the absolute quantification (copies/µL) was assessed as a merger of two replicates in bidimensional visualization applying the threshold based on the NTC signal. Samples were repeated when: (i) droplet number was below 10,000, (ii) there were too many positive droplets (i.e., higher than 50% of total droplets), (iii) the standard deviation was higher than 50 and/or higher than the mean of two replicates. The absolute quantification was obtained as ratio between the copy number of Exo-miRs and the UniSp6. The relative expression of miRs in the human NSCLC cells after transfection was assessed by qPCR. For each reaction, 60 ng of total RNA isolated by miRNeasy Kit, including a step of DNA digestion, was used. Each sample was amplified in duplicate on RealPlex2 system (Eppendorf) using iTaq Univer SYBR Green Supermix (Bio-Rad Laboratories) and normalized against U6 snRNA (Appendix A). The relative expression compared to the negative control (cells transfected with a scrambled control) was assessed by the formula 2^−∆∆CT^.

### 2.4. NanoString Gene Expression Analysis

Gene expression analysis was performed on transfected NSCLC cell lines using the nCounter platform (NanoString Technologies, Seattle, WA, USA) according to the manufacturer’s instructions. Specifically, we used the Human PanCancer Progression Panel that includes 740 cancer genes involved in the tumor progression processes such as angiogenesis, extracellular matrix remodeling, epithelial-to-mesenchymal transition and metastasis, plus 30 internal reference controls. Briefly, 150 ng of total RNA were bound with a Reporter CodeSet and a Capture ProbeSet and then hybridized at 65 °C for 21 h. Samples were purified and then loaded onto the nCounter Cartridge by the nCounter Prep Station, and RNA was quantified by using the Digital nCounter Nanostring. RCC files deriving from counting process were first evaluated using nSolver 4.0 Analysis Software (Nanostring) performing a quality control through the fields of view count, binding density parameter and the eventual presence of any warning flags. Data were then analyzed using the Advanced Analysis 2.0 plug-in of nSolver system. Target genes which achieved ≥1.5 and ≤−1.5-fold change values and *p*-value < 0.05 were considered for analysis. Raw data are available in GEO (ID: GSE198957).

### 2.5. Cell Culture, miR Transfection and Apoptosis/Proliferation Assays

The human A549 and NCI-H2172 (H2172) cell lines were obtained from Interlab Cell Line Collection (ICLC; IRCCS Ospedale Policlinico San Martino, Genova, Italy) and American Type Culture Collection (ATCC, Manassas, VA, USA), respectively. The short tandem repeat profiles were confirmed prior the start of any in vitro experiments. NSCLC cells were cultured in RPMI culture medium and 10% fetal bovine serum under the standard conditions with 5% CO2 atmosphere at 37 °C. The hsa-miR-130a-3p (YM00472237-ADA), hsa-miR-17-5p (YM00471966-ADA) and Negative Control (NC; YM00479903-ADA) miRCURY miRs were purchased from Qiagen. A549 and H2172 cells were seeded at 5 × 10^4^/mL and 10^5^, respectively, in 24-well plates and transfected with 50 nM miRCURY miR with lipofectamine 2000 (Thermo Fisher Scientific), according to manufacturer’s instructions. After 72 or 96 h, cells were seeded at 5 × 10^4^/mL in 24-well plates and transfected with 50 nM miRCURY miR with lipofectamine 2000 (Thermo Fisher Scientific), according to manufacturer’s instructions. After 72 h, cells were harvested to measure miR levels after transfection (qPCR) and evaluate proliferation (cell counts), apoptosis (Annexin V-FITC/PI Staining Assay) and cell cycle. For apoptosis detection, 1 × 10^5^ transfected cells were labeled with Annexin V-fluorescein isothiocyanate (FITC) and Propidium Iodide (PI) accordingly to manufacturer’s instructions (Thermo Fisher Scientific). Flow cytometer (FACSCalibur, Becton Dickinson, East Rutherford, NJ, USA) was utilized to acquire data and CellQuest (Becton Dickinson) software for data analysis. For the cell cycle analysis, 1 × 10^5^ transfected cells were incubated in PI staining solution. Cell cycle was determined using FACSCalibur (Becton Dickinson) and analyzed by ModFit (Becton Dickinson). For each analysis four biological replicates were performed.

### 2.6. Solid Phase Extraction and Atmospheric Pressure-Matrix Assisted Laser Desorption Ionization/Mass Spectrometry (AP-MALDI/MS) Analysis

The LMW protein fraction was extracted from serum samples as described in our previous study [24] using functionalized magnetic beads (DynabeadsVR RPC 18 Life Technologies Dynal, Irvine, CA, USA) with a C18 alkyl-modified surface. A synthetic internal standard peptide (MW 1419.76) was spiked (3.5 pmol/L) in the eluting solution to normalize the data. A standard procedure has been used to prepare the target plates for the AP-MALDI/MS analysis [25]. AP-MALDI analysis was acquired in positive mode on a Q-Exactive Plus Orbitrap mass spectrometer (Thermo Fisher Scientific) equipped with an AP-MALDI Ion Source (MassTech). We have applied the following parameters: spray voltage 3.8 kV, capillary temperature 320° C, S-lens RF level 100 and sheath and auxiliary gas flow rate were 3 and 1, respectively. High-resolution mass spectra, in the range from 800 to 3000 m/z, were acquired using a laser power at 35% of maximum, at resolution 70,000, autogain control 5e6 and maxIT 100ms. Full scan data were processed with Xcalibur v.4.1 (Thermo Fisher Scientific). Data acquisition was automated using Target NG 8.8.3 AP/MALDI PDF Control Software (MassTech) and programmed at a Laser Repetition Rate of 10,001 Hz. The irradiation program was automated using the spiral motion control of the PDF-MALDI ion source. Raw data were converted from the proprietary format (.RAW) to the mzML standard format by using MSconvert, a tool of the software suite ProteoWizard [26]. Single acquisition scans were obtained from the mzML formatted data by means of the MALDIquant R package [27]. Since each acquisition corresponded to five distinct aliquots of the same sample, five spectra, corresponding to the quintuplicated spots, were then determined for each sample by averaging 227 consecutive scans, corresponding to about 90 s of acquisition. In the resulting spectrum, two peaks may be considered separated if differing by at least 0.02 m/z. Software used for the identification of the starting and ending scans and for the computation of the spectra is available from the authors upon request by the interested parties. The integrity of each serum sample was tested by means of SeraDeg [28], while Geena 2 [29] was used for pre-processing of spectra which were then analyzed by using the SAM (Significance Analysis of Microarrays) statistical method in order to identify signals significantly different in the two groups (relapsed vs. non-relapsed patients). For each signal, the SAM analysis produced a *q-*value, described as a *p-*value adapted to a large number of comparisons, which was applied by using a threshold of 0.05 to assess the significance of the difference for that signal. The SAMR software that implements the SAM method by using the R statistical language was used to perform this task [30].

### 2.7. Statistical Analysis

Univariate and multivariable stepwise Cox regression analyses were carried out using both biological and clinical variables by applying the coxph function from the survival R package. For microarray data, prefiltering based on fold-change >1.5 between cases with and without progression was done before applying univariate Cox regression analysis. Akaike’s Information Criterion (AIC), evaluated using the AIC function from the stat R package, was used to compare models, where lower AIC means better fitting of the model. The risk score to select patients according to their association to relapse was calculated by means of a weighted sum of the variables in the model, where the weights are the Cox regression coefficients (Hazard Ratio: HR). Predictor values are centered using their overall means. Cell line analyses were performed using PRISM 9 (Graph-Pad Software, San Diego, CA, USA), applying the unpaired Student’s t-test.

## 3. Results

### 3.1. Patient Selection and Biological Sample Collection

Biological samples from 110 patients who had undergone resection between 2009 and 2012 were available at the CRB-HSM within the IRCCS Ospedale Policlinico San Martino (Genova, Italy). Among these, 43 patients were excluded as follows: (i) 24 with lung lesions that were actually metastases from other primary tumors, (ii) 6 with neuroendocrine differentiation tumors, (iii) 8 treated with neoadjuvant chemotherapy, (iv) 5 who died of another primary tumor during their follow-up (Figure 1). Hence, 67 patients were considered eligible for this study; the clinic-pathological characteristics are summarized in Table 1 and fully reported in Appendix A. The median age of patients was 68 years and the disease stages were classified as follows: stage I (42%), stage II (46%) and stage IIIA (12%). Among the 67 patients, 45 had lung adenocarcinoma and 22 had squamous cell carcinoma (one of whom had mixed adeno-squamous histology). Disease recurrence was reported in 9 patients (40.9%) with squamous cell carcinoma and 21 patients (46.7%) with adenocarcinoma. The median disease-free survival (DFS) and OS for patients who had disease recurrence were 22 and 32 months for squamous cell carcinoma and 18 and 57 months for adenocarcinoma, respectively, whereas the median DFS and OS for non-progressed patients were 91 and 102 months for squamous cell carcinoma and 88 and 90 months for adenocarcinoma, respectively.

### 3.2. Exo-miRNome Profiling by Microarray

The Exo-miRNome (intensity levels of 2549 miRs) was successfully analyzed in the plasma isolated before surgery from 67 non-metastatic NSCLC patients. In particular, the mean number of Exo-miRs in patients with a clinical event was higher than in cases without an event (213 vs. 171, *p*-value = 0.1) (Figure 2). Cox univariate regression analysis on the Exo-miRs exhibiting absolute fold-change higher than 1.5 in progressing cases vs. cases without event, yielded 9 miRs associated with DFS, all with positive HR and *p*-value less than 0.05 (Table 2).

### 3.3. Exo-miR Validation by ddPCR

The nine Exo-miRs identified by the microarray as predictive markers of DFS were initially validated by an independent digital tool. Specifically, validation was performed in 63 out of 67 patients by ddPCR (Appendix A). All the nine Exo-miRs showed positive Spearman correlation coefficients between microarray and ddPCR values, with *p*-value lower than 0.01 (Appendix A and Appendix A); however, the expression levels of Exo-let-7i-5p in ddPCR showed an opposite trend compared with the microarray data, with a higher median value in the event-free patient cohort, and was consequently excluded from the subsequent analysis (Appendix A). The univariate Cox regression HR and *p*-values of the eight miRs are reported in Table 3.

Contrarily to microarray data, Exo-miR-130a-3p was the only miR significantly associated with increased HR (worse prognosis) (*p*-value = 0.005) (Appendix A). Then, multivariable stepwise Cox regression analysis using the eight miRs yielded a combination of Exo-miR-130a-3p and Exo-miR-17-5p as best associated with DFS. In the univariate regression, Exo-miR-17-5p reported a non-significant coefficient HR slightly higher than 1 (HR = 1.028; *p*-value = 0.623; Table 3), while in the combined score it reported a significant coefficient lower than 1 (HR = 0.6748; *p*-value = 1.87 × 10^−5^) that reduced the strong effect of Exo-miR-130a-3p (Table 4).

When clinical information such as age and stage was added, the algorithm selected age as a predictive factor for progression, together with Exo-miR-130a-3p and Exo-miR-17-5p, providing a model that performs better (‘miR-130a-3p+miR-17-5p’ Akaike Information Criterion (AIC): 239.09 vs. 236.37 ‘miR-130a-3p+miR-17-5p+Age’; Table 4). Kaplan–Meier curves stratified by median level of the two combination scores, without and with age, are shown in Figure 3A,B.

### 3.4. Peptidome Analysis

We initially analyzed by AP-MALDI/MS the LMW protein profile of 67 serum samples (quintuplicate). In order to avoid procedural biases, the samples were randomly analyzed. The SeraDeg analysis selected a set of samples of adequate quality and preservation including 47 pathological samples, of which 26 were from relapsed and 21 from non-relapsed patients (Appendix A). The selected sample set was used to identify possible candidate peptides able to distinguish patients with relapse. Geena 2 was used to carry out the pre-processing steps and a total of 862 peaks were detected. Finally, the number of useful signals was reduced to 393 after visual inspection and removal of sporadic signals. By comparing the two groups of samples, the SAM analysis highlighted four signals (1206.56, 1263.58, 1350.62 and 1465.64 m/z) having a *q*-value of 5% or less in the comparison between relapsed and non-relapsed patients (Table 5; Figure 4). In particular, these signals (1206.56, 1263.58, 1350.62 and 1465.64 m/z) were significantly lower in relapsed patients (fold change 0.66, 0.31, 0.53 and 0.50, respectively) (Table 5).

The results of the peptides from the univariate Cox regression analysis are shown in Table 6 and Appendix A. Notably, these peptides originate from fibrinogen alpha chain generated during the coagulation process. Consequently, we evaluated the correlations among the Fibrinopeptide A (FpA)-derived fragments and some coagulation parameters (i.e., Prothrombin time, International Normalized Ratio, Partial Thromboplastin Time, Fibrinogen Antithrombin III), but no significant association was found (*p*-value > 0.09) (Appendix A).

### 3.5. Combination of Exo-miRs and Peptides

When peptides were combined with Exo-miRs for 46 of 47 patients (for one patient the ddPCR Exo-miR value was unavailable), the Cox regression selected only one FpA-derived fragment (Mean 1465.64 m/z, greater) (Table 7). Among the Exo-miRs, the model selected both Exo-miR-130a-3p and Exo-miR-17-5p, the latter showing the same mitigation behavior compared to miR-130a-3p; however, Exo-miR-17-5p again has HR < 1 (HR = 0.7386) compared with its value in the univariate Cox regression (HR = 1.040) (Table 6). When we excluded the Exo-miR-17-5p, although the AIC value reported a small increase (152.68 vs. 144.48 with three variables), the score was still able to discriminate high-risk patients (Table 7; Figure 5A,B). In the analysis including peptidome data, no clinical variable was selected, probably due to fewer patients or stronger association of peptides with DFS.

### 3.6. Gene Expression Analysis in Transfected Cells

To investigate the potential role of Exo-miRs in cancer progression, we analyzed the effect of miR-130a-3p as well as miR-17-5p in an in vitro model by transfecting two human NSCLC cells (A549 and H2172) with specific miR mimics. The expression patterns of 740 cancer progression genes in transfected cells were studied using the nCounter platform. Four independent biological experiments for each transfection (“miR-130a-3p”, “miR-17-5p”) were profiled and compared to the negative controls (four replicates of cells transfected with the negative control). All transfections significantly increased the expression of both miRs ranging from 461 to more than 30,000-fold in an experiment-dependent manner (Figure 6).

Gene expression analyses showed generally more genes deregulated in the H2172 cells (“miR-130a-3p”: 59 down-, 118 up-; “miR-17-5p”: 143 down- and 53 up-modulated) than the A549 cells (“miR-130a-3p”: 28 down-, 30 up-; “miR-17-5p”: 79 down-, 27 up-modulated) (Appendix A). Furthermore, a relatively low number of deregulated genes were shared between the two transfected cell lines with the exception of the down-modulated genes in A549 transfected with miR-17-5p, which reported about 60% (47/79) of genes in common with H2172 (Figure 7C, Table 8).

Despite these differences, the transfections with miR-17-5p mimic silenced the most relevant cancer progression pathways such as cell proliferation, TGF Beta signal, angiogenesis and cell adhesion genes in both cell lines (Figure 8; Appendix A).

Furthermore, while the A549 cell line showed no significant changes in cell proliferation, death rates or cell cycle following miR-17-5p transfection at 72 h, H2172 cells behaved differently. Indeed miR-17-5p transfection altered H2172 cell apoptosis with a weak increase in early apoptotic cells and a significant decrease in G0/G1 phase (G0/G1 phase % mean: 45.0% vs. 51.7% control; *p*-value = 0.03), associated with a higher percentage of S-phase cells (S phase % mean: 41.8% vs. 35.9% control; *p*-value = 0.037) (Figure 9A–C). This effect was also confirmed at 96 h after miR-17-5p transfection, with a further increase in early apoptotic cells (% mean: 11.5% vs. 7.8% control; *p*-value = 0.009) without significant changes in cell counts (Figure 9D–F).

In contrast to the miR-17-5p, the transfection with miR-130a-3p generally reported a higher number of up-regulated genes particularly in the H2172 cells (Figure 7; Appendix A). Notably, the pathway analysis showed activations in the H2172 cells of several signaling linked to tumor progression such as cell proliferation, metastatic processes, epithelial mesenchymal transition (EMT) and angiogenesis (Figure 10; Appendix A). A similar trend was also found in the transfected A549 cells which reported an activation of the cell proliferation, angiogenesis and blood coagulation (Figure 10).

However, no significant changes were found in cell proliferation, cell death rates, or cell cycle in any of the cells after its transfection (Figure 11A–C).

## 4. Discussion

The early detection of poor outcomes in patients with resected lung cancer represents the key to successful clinical management of this population of patients. To date, beyond disease stage at surgery, extremely limited predictive biomarkers have been identified in the post-operative setting [6]. Furthermore, the adjuvant treatment for early-stage NSCLC has recently been updated due to the introduction of an immune checkpoint blockade [31]; hence, improving the identification of patients with high risk of recurrence after resection might result in the optimization of adjuvant strategies for NSCLC. Growing evidence reports that tumor progression might also depend on a dual interconnection between cancer cells and the surrounding TME [7]. The circulating exosome, entrapping genetic material such as miRs, has been demonstrated to be a potent mediator in the crosstalk between tumor and microenvironmental cells such as stromal and immune cells [10]. Specific peptidome patterns have also been correlated with cancer patient outcomes [12]. In the present study, we performed an extensive analysis of circulome-derived markers, including Exo-miRnome and peptidome, to build a prognostic score that turned out to be an ideal model. The Exo-miR profiling showed a significant increase in Exo-miR-130a-3p in patients with relapse. MiR-130a-3p has mainly been described as down-modulated in lung cancer compared to normal lung tissue [32], but the Exo associated form has been correlated with an unfavorable prognosis in some cancer types [33,34]. This opposite effect is not surprising; indeed, Exo-miRs, as being circulating, can act not only on tumor cells but on immune and stromal cells as well, modifying their functions in a context-dependent manner. Notably, multivariable analysis yielded a combination of the Exo-miR-130a-3p with Exo-miR-17-5p as best model associated with DFS. Exo-miR-17-5p alone showed a slightly significant positive association with DFS by microarray data (HR = 1.253, *p*-value = 0.023, n = 67) but it was not associated with DFS by ddPCR data (HR = 1.028, *p*-value = 0.623, n = 63); however, when combined in a multivariable model, the HR of Exo-miR-17-5p became lower than “one” (i.e., negatively associated with DFS) and significant, underlying an interaction with miR-130a-3p that seems to balance its positive contribution to the final predictive score. The modification in HR value has already been documented previously [35], and this event might be related to the low number of patients as well as to the small contribution of Exo-miR-17-5p in the model compared to Exo-miR-130a- 3p. Indeed, in the univariate analysis of 46 patients, the individual contribution of miR-17-5p was limited, as its HR was close to “one” and statistical significance was not reached (HR = 1.040; *p*-value = 0.561).

Hence, in order to understand the role of the two miRs in lung cancer progression, we also investigated their effects in the in vitro model. We generally found a silencing of the major pathways involved in cancer progression after miR-17-5p transfection, with also a significant increase in early apoptosis, particularly in the H2172 cell line. The absence of this effect in A549 cells could be due to the activating mutation in the *KRAS* gene [36], which in turn could limit the tumor suppressor effect of miR-17-5p, independently activating cell proliferation. We also observed an increase in S-phase cells after miR-17-5p transfection. Similar modifications in cell cycle profile have already been described after miR-17-5p overexpression [37]. In particular, the authors demonstrated that miR-17-5p was a key regulator of the G1/S phase transition acting both as an oncogene and as a tumor suppressor in different cellular settings. These data lead to hypothesize a main suppressive role of the miR in this context. Conversely, miR-130a-3p transfection showed activation of some crucial signaling pathways for tumor development. Among the commonly deregulated genes, we found a significant up-regulation of the Rac Family Small GTPase 1 (*RAC1*) gene and neuropilin 2 (*NRP2*), which are involved in tumor migration and invasion in different tumors [38] including lung cancer [39,40]. In particular, Rac1 protein participates in the membrane protrusion formation, driving tumor cell migration [41] and the formation of membrane ruffles [42], also implicated in the metastatic potential of tumor cells [43]. In addition, we also found deregulations of TME signaling involved in the onset and progression of cancer. Among these, angiogenesis was predicted in the miR-130a-3p transfections. Currently, a number of studies support the role of miR-130a-3p in angiogenesis activation [44,45,46]. Among the commonly up-regulated genes after miR-130a-3p, we detected a significant increase in neuropilin 2 (*NRP2*) involved in the formation of new blood vessels [47]. In particular, very recently, Alghamdi and colleagues demonstrated that the NRP2 by RAC1 protein promote the adhesion and migration of the endothelial cells [48]. Additionally, we also observed an up-regulation of the Integrin Subunit Beta 1 (*ITGB1*), a protein involved in the vasculogenic mimicry, an alternative tumor mechanism of vessel-like structure formation [49].

The serum peptidome analysis evidenced a lower abundance of four peptides (i.e., 1206.56, 1263.58, 1350.62 and 1465.64 m/z) in progressed patients and these peaks matched with the FpA. However, the presence of degradation fragments of fibrinogen was also used as a quality control (SeraDeg) in the selection of serum specimens, but in our control, low-quality samples are characterized by a reduction in FpA-derived fragments of greater mass (1350.62, 1465.64 m/z) and abundance in smaller ones (905.47, 1077.52 m/z) [28]. In contrast, in our progressed patient cohort we observed a reduction in high mass spikes only, suggesting a cancer pattern. When we combined the peptides to Exo-miRs and clinical data in the multivariable analysis, the fragments improved the risk scores. Notably, the greater FpA-derived fragment with mean 1465.64 m/z (FpA(2–16)) was the only one retained by step-wise regression analysis. A number of serum peptidome studies have described distinctive FpA patterns associated to clinically-relevant outcomes; however, their biological significance in the blood of cancer patients is still debated. Although FpA-derived fragments were mainly found overexpressed in cancer patients [18,50,51,52,53], a downmodulation has also been associated with several cancer conditions [52,54]. In 2006 Villanueva et al. published a study in which lower levels of FpA-derived fragments were found in serum from patients affected by prostate, bladder and breast cancer [12]. Interestingly, the same authors found similar results in metastatic thyroid carcinoma supporting the hypothesis that low levels of FpA may be related to advanced cancer conditions [13]. A possible explanation of our findings might be linked to a lower concentration of fibrinogen in the serum of relapsed patients, although no correlation was found in our patient cohort. Several tumor-related disorders of hemostasis have been described, including the fibrinogen fragmentation [55,56]. Furthermore, since serum peptidomes derive mainly from blood components, they particularly reflect proteolytic degradation products and can therefore be considered as indicators of endopeptidase activity in plasma. In this regard, it has also been reported how tumor cells can produce particular exopeptidases [12], linked to the FpA tumor pattern [54]. In light of these studies, we can hypothesize that depletion of FpA fragments in patients with progression may be the result of tumor-derived protease activation. In support of this, in both transfected cells we found a significant increase in metalloprotease (i.e., metallopeptidase domain 17, *ADAM17*) and coagulation process such as *RAC1* gene, already described as a potent clotting activator in vitro [57].

In spite of the intriguing results of our study, some potential limitations need to be considered. First, the patient cohort is relatively small and there is no validation in an independent population to assess the robustness of the generated risk score. Second, the lack of serum samples from healthy individuals collected simultaneously with patient serum and longitudinal sampling in the cancer cohort hampered definitive conclusions on the peptidome. Furthermore, clinical parameters such as coagulation status of patients were available for a limited selection of patients, and a fibrinolysis parameter (e.g., d-dimer) was lacking, which prevented us from understanding the role of coagulation in these patients. Last but not least, although numerous studies have already shown a connection between miR-130a-3p and angiogenesis, we have not performed in vitro studies proving its role in remodeling the TME.

## 5. Conclusions

To the best of our knowledge this is the first study that reports a combination of Exo-miR and peptidome to identify patients with NSCLC at higher risk of recurrence after surgery for curative intent. Notably, we found that the combination of Exo-miR-130a-3p with FpA (2-16) is strongly associated with DFS and is able to identify a group of high-risk patients with median DFS of 18 months compared to the low-risk group (median not reached). Together, these results support the proof-of-concept for integrating circulating markers to significantly improve patient’s risk score. In addition, mimic transfections have shown that higher levels of miR-130a-3p can cause a deregulation of some gene pathways involved in metastasis and angiogenesis suggesting a potential role of this miR in cancer progression. Furthermore, the gene enrichment in coagulation and metalloprotease could explain the FpA reduction in patients with cancer progression. In conclusion, the identification of patients at high risk of relapse certainly has a relevant impact in the management of patients with resectable NSCLC, as these data could potentially assist to allocate such patients to adjuvant therapies and post-surgical, more intensive instrumental follow-ups. However, in the perspective to transfer these results into a clinical setting, they will need to be validated in larger series of patients.

## Figures and Tables

**Figure 1 cancers-14-03412-f001:**
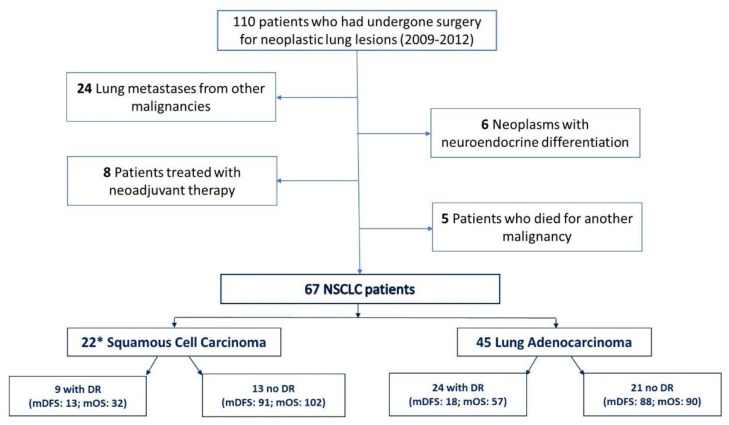
Flow chart of patient selection based on eligibility criteria. * One patient had adeno-squamous histology. Abbreviation: DR: disease recurrence; mDFS: median disease-free survival in months; mOS: median overall survival in months.

**Figure 2 cancers-14-03412-f002:**
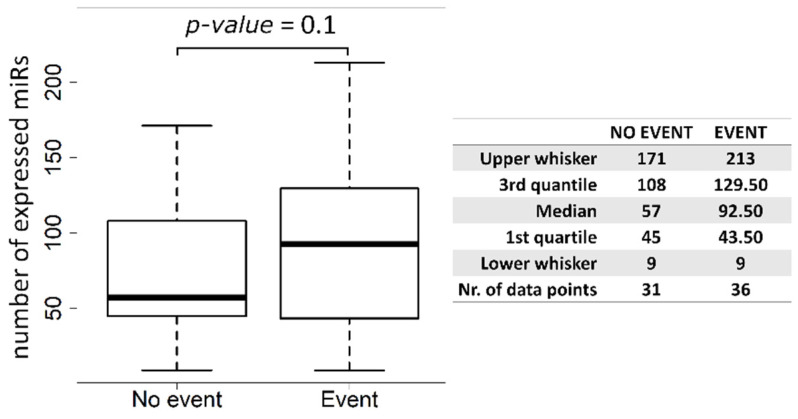
Box plots of number of expressed Exo-miRs in plasma samples from NSCLC patient without event (No event) versus patients with a progression of disease (Event).

**Figure 3 cancers-14-03412-f003:**
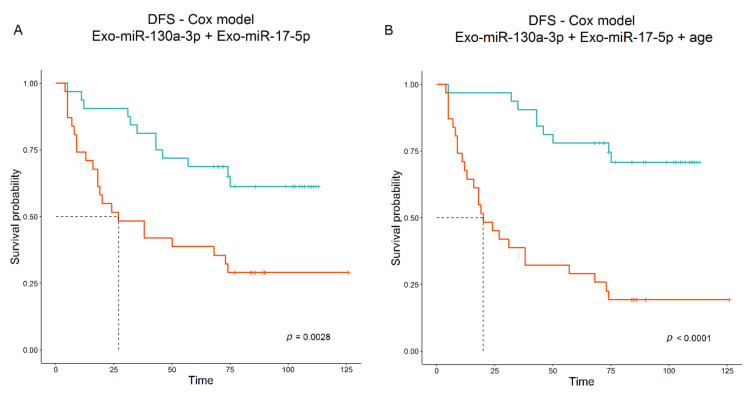
(**A**) Kaplan–Meier curves obtained by stratifying 63 patients according to the model score that combines Exo-miR-130a-3p and Exo-miR-17-5p (ddPCR data). Median DFS is 27 months. (**B**) Kaplan–Meier curves obtained by stratifying 63 patients according to the model score that combines Exo-miR-130a-3p, Exo-miR-17-5p (ddPCR data) and age. Median DFS is 20 months.

**Figure 4 cancers-14-03412-f004:**
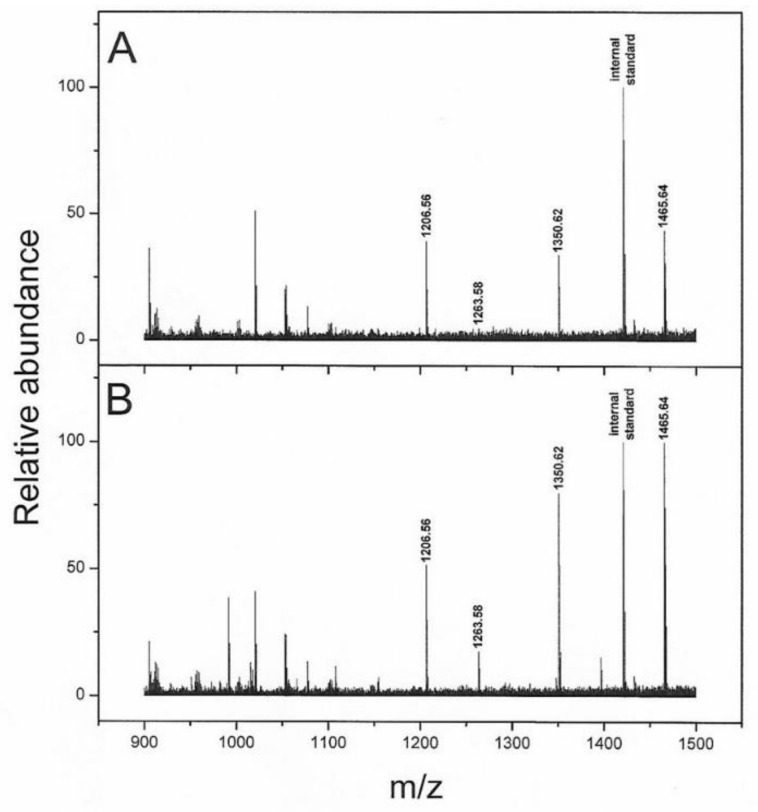
Example of MS spectra. MALDI/MS spectra of serum samples obtained from recurrent (**A**) and non-recurrent (**B**) NSCLC patients. The FpA-derived fragments and the internal standard (at 1420.76 m/z) are labelled with mono-isotopic m/z value. The more preserved forms of FpA are reduced in the serum sample from recurrent patient. Abbreviation: is: Internal Standard.

**Figure 5 cancers-14-03412-f005:**
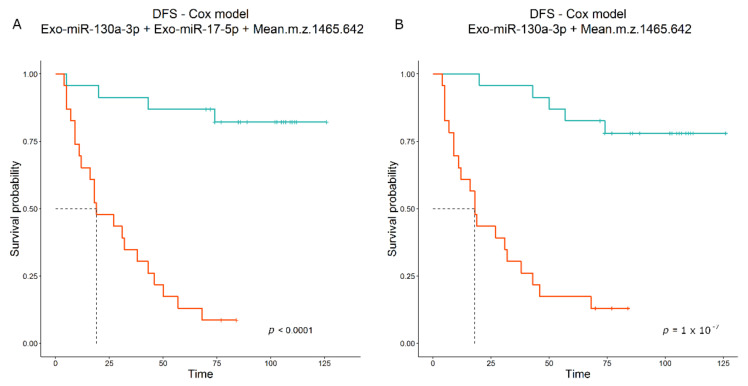
(**A**) Kaplan–Meier curves obtained by stratifying 46 patients according to the model score that combines Exo-miR-130a-3p, Exo-miR-17-5p (ddPCR data) with 1465 m/z (FpA(2–16)). Median DFS is 19 months. (**B**) Kaplan–Meier curves obtained by stratifying 46 patients according to the model score that combines Exo-miR-130a-3p and 1465 m/z (FpA(2–16)). Median DFS is 18 months.

**Figure 6 cancers-14-03412-f006:**
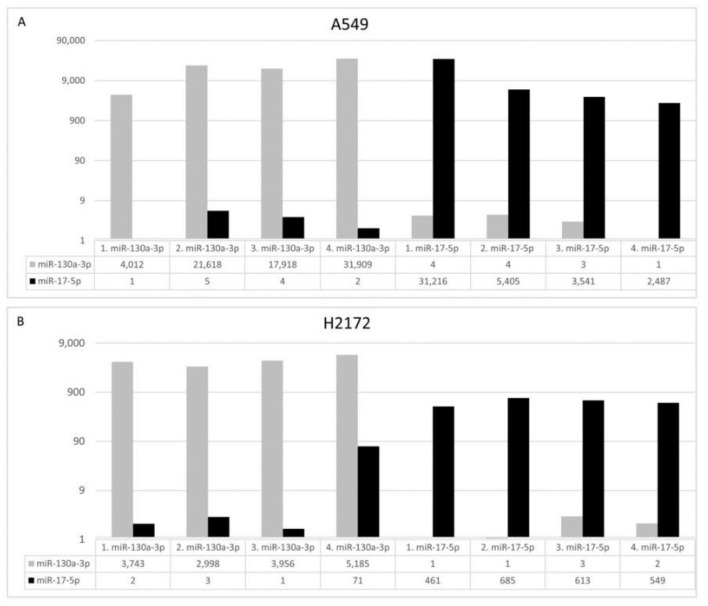
Bar charts report the fold changes of miR-130a-3p (grey) and miR-17-5p (black) at 72 h after their transfections in A549 (**A**) and H2172 (**B**) cells. In each experiment the miR transfected cells were compared to the cells transfected with a negative control. *X*-axis: reports the transfections with the fold changes; *Y*-axis: reports the fold values transformed in base 10-log scale.

**Figure 7 cancers-14-03412-f007:**
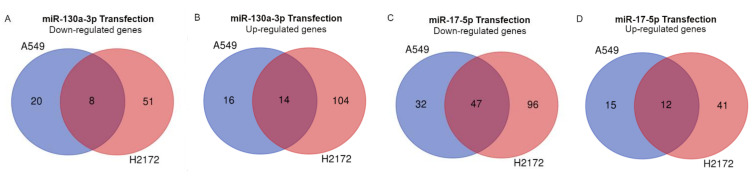
Venn Graphs show the deregulated genes shared by two cells after the transfections with miR-130a-3p (**A**,**B**) and miR-17-5p (**C**,**D**).

**Figure 8 cancers-14-03412-f008:**
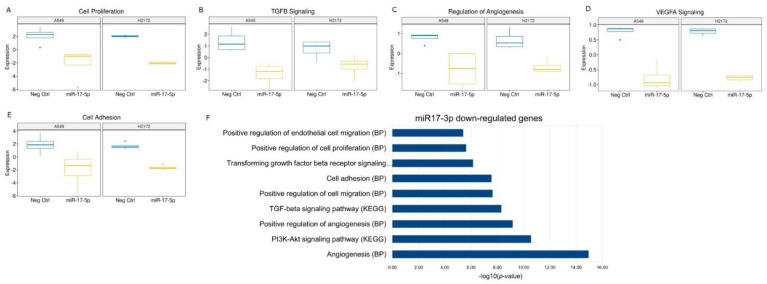
(**A**–**E**) Box plots of the enriched pathways predicted by gene expression values on nSolver software analysis of the specific pathway down-regulated in the transfected cells with the miR-17-5p. (**F**) Bar chart reports the most relevant functional annotations obtained by DAVID Bioinformatics tool using down-modulated genes in miR-17-5p transfected cells versus NC. The list was sorted by log-false discovery rate (logFDR) values. BP: Biological Process.

**Figure 9 cancers-14-03412-f009:**
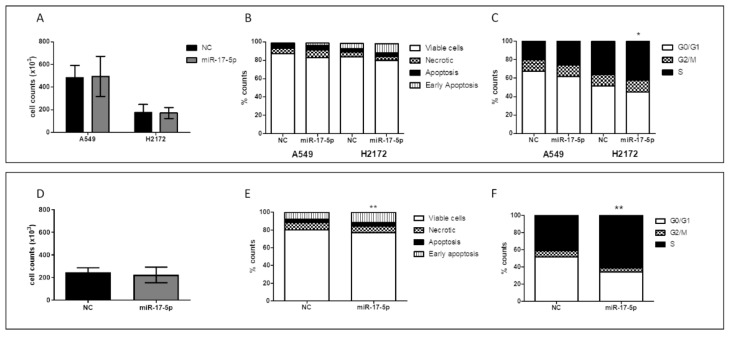
(**A**) Cell counts, (**B**) apoptosis and (**C**) cell cycle of A549 and H2172 cell lines after transfection with miR-17-5p or scrambled control (NC) at 72 h. (**D**) Cell counts, (**E**) apoptosis and (**F**) cell cycle of H2172 NSCLC cell line after transfection with miR-17-5p or NC at 96 h. Cell counts are expressed as the mean ± standard deviations (four experiments). Stacked plots are the mean of four different experiments. Standard deviation was below 5% of the mean. * *p*-value < 0.05; ** *p*-value < 0.005.

**Figure 10 cancers-14-03412-f010:**
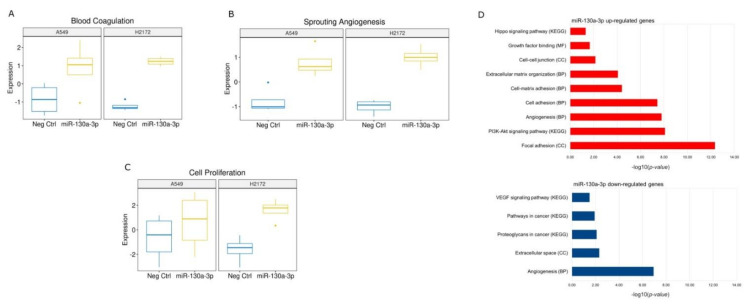
(**A**–**C**) Box plots of the enriched pathways predicted by gene expression values on nSolver software analysis of the specific pathway up-regulated in the transfected cells with miR-130a-3p. (**D**) Bar charts report the most relevant functional annotations obtained by DAVID Bioinformatics tool using the overexpressed (red bar) and down-modulated (blue bar) genes in miR-130a-3p transfected cells versus NC. The lists were sorted by log-false discovery rate (logFDR) values. BP: Biological Process; MF: Molecular Function; CC: Cellular Component.

**Figure 11 cancers-14-03412-f011:**
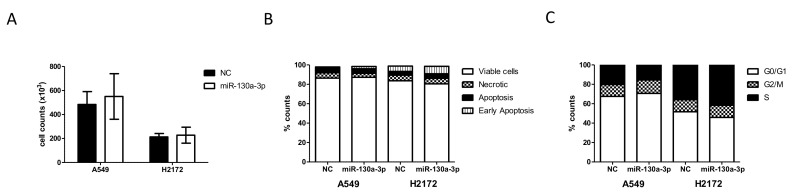
(**A**) Cell counts, (**B**) apoptosis and (**C**) cell cycle of A549 and H2172 NSCLC cell lines after transfection with miR-130a-3p or scrambled control (NC) at 72 h. Cell counts are expressed as the mean ± standard deviations (four experiments). Stacked plots are the mean of four different experiments. Standard deviation was below 5% of the mean.

**Table 1 cancers-14-03412-t001:** Clinical and pathological characteristics of the 67 eligible patients.

Characteristics	n. (%)	%	*p*-Value		HR	
			DFS	OS	DFS	OS
**Median age (range)**	68 (47–84)		0.17	0.14	1.04	1.04
**Gender**						
Male	51	76.1	0.20	0.14	1.89	2.09
Female	16	23.9	0.20	0.14	0.53	0.48
**Histotype**						
Adenocarcinoma	45	67.2	0.87	0.76	1.07	1.13
Squamous cell carcinoma *	22	32.8	0.87	0.76	0.94	0.88
**ECOG PS**						
0	23	34.3	0.06	0.05	0.33	0.30
≥1	14	20.9	0.06	0.05	3.00	3.35
Missing data	30	44.8				
**Smoking habit**						
Never smoker	2	3.0				
Former smoker	25	37.3	0.50	0.91	0.49	0.89
Smoker	23	34.3	0.53	0.98	0.51	0.97
Missing data	17	25.4				
**Stage**						
Stage I (A+B)	28	41.8				
Stage II (A+B)	31	46.3	0.66	0.49	1.19	1.32
Stage III A	8	11.9	0.13	0.17	2.28	2.10
**Adjuvant Chemotherapy**						
Yes	21	31.3	0.44	0.44	0.73	0.73
No	46	68.7	0.44	0.44	1.38	1.37

* One patient had adeno-squamous histology.

**Table 2 cancers-14-03412-t002:** Coefficients and relative *p*-value of the univariate Cox regression models using 9 Exo-miRs, obtained by microarray (n = 67).

Exo-miRs	HR	*p*-Value
*Exo-let7a-5p*	1.241	0.020
*Exo-let7b-5p*	1.325	0.018
*Exo-let7f-5p*	1.217	0.025
*Exo-let-7i-5p*	1.259	0.025
*Exo-mir-103a-3p*	1.240	0.018
*Exo-mir-130a-3p*	1.268	0.009
*Exo-miR-17-5p*	1.253	0.023
*Exo-miR-24-3p*	1.240	0.023
*Exo-mir-30b-5p*	1.286	0.017

**Table 3 cancers-14-03412-t003:** Coefficients and relative *p*-values of the univariate Cox regression model using 8 Exo-miRs on ddPCR data (n = 63). In bold is reported significant *p*-value (<0.05).

Exo-miRs	HR	*p*-Value
*Exo-let7a-5p*	1.038	0.476
*Exo-let7b-5p*	1.027	0.697
*Exo-let7f-5p*	1.056	0.290
*Exo-mir-103a-3p*	1.057	0.323
*Exo-mir-130a-3p*	1.145	**0.005**
*Exo-miR-17-5p*	1.028	0.623
*Exo-miR-24-3p*	1.021	0.673
*Exo-mir-30b-5p*	1.084	0.122

**Table 4 cancers-14-03412-t004:** Coefficients and relative *p*-values of the stepwise multivariable Cox regression model obtained from the combination of 8 Exo-miRs obtained by ddPCR (n = 63), without and with clinical variables. AIC (Akaike Information Criterion) values of the models are reported.

Variables	DFS (Stepwise)		DFS (Stepwise)	
	HR	*p*-Value	HR	*p*-Value
*Exo-miR-130a-3p*	1.562	0.00000084	1.590	0.0000004
*Exo-miR-17-5p*	0.675	0.00001870	0.677	0.0000178
*Age*	-	-	1.040	0.0353000
**AIC**	239.097		236.369	

**Table 5 cancers-14-03412-t005:** Results of the SAM analysis for the comparison between relapsed and non-relapsed patients. Only significant signals having *q*-value < 5%, are reported. The SAM Score corresponds to a t statistic. Columns Num and Den report the values of the numerator and denominator of the fraction calculated by the method for determining the Score. A negative Score, as well as a Fold change < 1, highlights a decrease in the values in the second group.

Mass	Score	Num	Den	Fold Change	*q*-Value	Peptide	Sequence
*Mean 1206.56 m/z*	−3.45	−31.72	9.18	0.66	0.0000	FpA(5–16)	EGDFLAEGGGVR
*Mean 1263.58 m/z*	−3.18	−9.14	2.88	0.31	0.0000	FpA(4–16)	GEGDFLAEGGGVR
*Mean 1350.62 m/z*	−3.77	−34.50	9.16	0.53	0.0000	FpA(3–16)	SGEGDFLAEGGGVR
*Mean 1465.64 m/z*	−4.05	−66.91	16.51	0.5	0.0000	FpA(2–16)	DSGEGDFLAEGGGVR

**Table 6 cancers-14-03412-t006:** Coefficients and relative *p*-values of the univariate Cox regression model using 4 peptides.

Circulating Markers	HR	*p*-Value
*Mean m/z 1206.56*	0.984	0.003
*Mean m/z 1350.62*	0.982	0.002
*Mean m/z 1465.64*	0.988	0.001
*Mean m/z 1263-58*	0.930	0.004

**Table 7 cancers-14-03412-t007:** Coefficients and relative *p*-value of the stepwise Cox regression analysis starting from 8 Exo-miRs (ddPCR data) combined with 4 peptides, or from Exo-miR-130a-3p alone combined with 4 peptides (n = 46). AIC (Akaike Information Criterion) values of the models are reported.

Variables	Exo-miR130a-3p+Exo-miR-17-5p+Mean 1465.64 m/z		Exo-miR130a-3p+Mean 1465.64 m/z	
	DFS (Stepwise)		DFS (Stepwise)	
	HR	*p*-Value	HR	*p*-Value
*Exo-miR-130a-3p*	1.566	0.00001	1.213	0.00105
*Exo-miR-17-5p*	0.739	0.00102	-	-
*Mean 1465.64 m/z*	0.989	0.00123	0.988	0.00049
**AIC**	144.480		152.680	

**Table 8 cancers-14-03412-t008:** List of the deregulated genes shared by two cell lines after transfections.

		Common Gene Number	Common Genes
miR-130a-3p	Down	8	*COL1A1, DST, PKN1, SKP1, CDC42, SOD1, CYB561, SRPK2*
	Up	14	*RAC1, AREG, ITGB1, ADAM17, NRP2, QKI, SET, PPP2CB, FGFR3, BMPR2, NOTCH1, HSD17B12, MTA1, F11R*
miR-17-5p	Down	47	*MCAM, PLAU, FGF2, CLU, C1S, RNH1, MAP2K4, APC, TGFBR2, DENND5A, GPX1, ITGB1BP1, MAP3K7, BNC2, ACVR1, CD24, NRP1, TIMP1, CYB561, BMP5, SYNE1, AKT1, AKT3, STAT3, SH2B3, HEG1, RHOA, BICC1, SPP1, ADAM9, AGGF1, RB1, PLCG1, ECM1, SIRT1, TIMP2, PKN1, TNFSF12, MYLK, BAD, SOD1, WIPF1, PIK3CA, CEP170, ADD1, RRAS, SRPK2*
	Up	12	*ENO3, SET, PPP2CB, FGFR3, RAC1, HSD17B12, FGFR4, PECAM1, RBM47 ITGB1, HDAC5, QKI*

## Data Availability

Data supporting our findings are included in the Appendix A, while raw data from microarray and NanoString are deposited in NCBI’s Gene Expression Omnibus accessible through GEO Series accession number: GSE198959 (1 May 2022), and peptidome analysis are available in MassIVE, at doi.org/10.25345/C5QZ22M3S, reference number MSV000089072.

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
