# Peer review of "A Circulating Risk Score, Based on Combined Expression of Exo-miR-130a-3p and Fibrinopeptide A, as Predictive Biomarker of Relapse in Resectable Non-Small Cell Lung Cancer Patients"

_cancers, 2022, doi:10.3390/cancers14143412_

Round 1
Reviewer 1 Report
In the current article, " A Circulating Risk Score, Based on Combination of Exo-miR-130a-3p and Fibrinopeptide A, as Predictive Biomarker of Relapse in Early Stage Non-Small Cell Lung Cancer Patients" the authors have discussed the possibility of Ex0-miR-130a-3p as a prognostic marker for relapse in NSCLC patients. I believe, the current manuscript can be accepted after addressing the following comments.
Comments
Comment 1: Kindly provide figure legends for supplementary figures.
Comment 2: Authors transfected A549 cells with miR-130a-3p or miR-17-5p for initial in vitro experiments. However, it will be beneficial to replicate similar experiments with other NSCLCs especially adenocarcinoma cell lines as well.
Comment 3: Are the levels achieved by transfection of A549 cells with miRs comparable to what is seen in NSCLC patient samples?
Comment 4: Authors discussed, a slight decrease in proliferation of CD3/CD28 stimulated PBMCs from healthy donors. However, the results shown in the supplementary figure do not look convincing enough to warrant that statement.
Comment 5: It will be crucial to characterize changes in gene signatures not only in transfected A549 cells but also in PBMCs following co-culture with Exo-miR-130a-3p and miR17-5p. This experiment will certainly help in characterizing the immune-inhibitory role of miRs.
Comment 6: One major drawback of the current study is that patients who received neoadjuvant chemotherapy are not selected. Is there any specific reason to exclude these patients? Normally, neo-adjuvant therapy is commonly the first line of treatment for NSCLC patients along with surgery.
Author Response
Reviewer #1:
In the current article, "A Circulating Risk Score, Based on Combination of Exo-miR-130a-3p and Fibrinopeptide A, as Predictive Biomarker of Relapse in Early Stage Non-Small Cell Lung Cancer Patients" the authors have discussed the possibility of Ex0-miR-130a-3p as a prognostic marker for relapse in NSCLC patients. I believe, the current manuscript can be accepted after addressing the following comments.
Dear reviewer#1 thank you for your suggestions; hereby our reply to the revision:
- Kindly provide figure legends for supplementary figures.
Author response: We apologize for this missing information; therefore, we have included the legends of the Supplementary figures in the Supplementary Materials (Lines:804-809).
- Authors transfected A549 cells with miR-130a-3p or miR-17-5p for initial in vitro experiments. However, it will be beneficial to replicate similar experiments with other NSCLCs especially adenocarcinoma cell lines as well.
Author response: We thank the reviewer for this helpful observation. Indeed, before starting our in vitro study, we evaluated the expression levels of both miRs in a panel of human lung adenocarcinoma cell lines including A549, NCI-H2342, NCI-H2172. In general, the miR expressions were quite similar among the cells; consequently, we have chosen A549 cell, because it reports a very high proliferation and is easier to keep in culture and transfect. However, we agree with the reviewer that data generated from a single cell line may not be representative and therefore the data generated should be interpreted with caution. Accordingly, we have investigated the role of miR-130a-3p or miR-17-5p using another adenocarcinoma cell line (i.e., NCI-H2172); the NCI-H2342 line was excluded for the difficult handling. Similar to the A549, this cell reported an increase of both miRs after mimic transfections (ranging from 461 to over 5,000; Figure 6B). Gene expression analyzes showed generally more genes deregulated in the H2172 cells ("miR-130a-3p": 59 down-, 118 up-; "miR-17-5p": 143 down- and 53 up-modulated) than the A549 cells ("miR-130a-3p": 28 down-, 30 up-; "miR-17-5p": 79 down-, 27 up-modulated) (Figure 7; Table S6-7). Furthermore, a relatively low number of deregulated genes were shared between the two transfected cell lines with the exception of the down-modulated genes in A549 transfected with miR-17-5p, which reported about 60% (47/79) of genes in common with H2172 (Figure 7 C; Table 8). Despite these differences, the transfections with miR-17-5p mimic silenced the most relevant cancer progression pathways such as cell proliferation, TGF Beta signal, angiogenesis and cell adhesion genes in both cell lines (Figure 8 A-E). Furthermore, while the A549 cell line showed no changes in cell proliferation, death rates, or cell cycle after miR-17-5p transfection at 72 hours, H2172 cells behaved differently. Indeed miR-17-5p transfection altered H2172 cell apoptosis with a weak increase in early apoptotic cells and a significant decrease in G0/G1 phase (G0/G1 phase % mean: 45.0% vs. 51.7% Control; p-value = 0.03), associated with a higher percentage of S-phase cells (S phase % mean: 41.8% vs. 35.9% Control; p-value = 0.037) (Figure 9 A-C). This effect was also confirmed at 96 hours after miR-17-5p transfection, with a further increase in early apoptotic cells (% mean: 11.5% vs. 7.8% Control; p-value = 0.009), without significant changes in cell counts (Figure 9 D-F). The absence of this effect in A549 cells could be due to the activating mutation in the KRAS gene [36], which in turn could limit the tumor suppressor effect of miR-17-5p, independently activating cell proliferation. We also observed an increase in S-phase cells after miR-17-5p transfection. Similar modifications in cell cycle profile have already been described after miR-17-5p overexpression [37]. In particular, the authors demonstrated as miR-17-5p was a key regulator of the G1/S phase transition acting both as an oncogene and as a tumor suppressor in different cellular settings. These data lead to hypothesize a main suppressive role of the miR in this context.
In contrast to the miR-17-5p, the transfections with miR-130a-3p generally reported a higher number of up-regulated genes, particularly in the H2172 cells (Figure 7; Table S6-7). Notably, the pathway analysis showed activations of several signaling pathways linked to tumor progression such as cell proliferation, metastatic processes, epithelial mesenchymal transition (EMT), and angiogenesis in the H2172 (Figure 10; Table S8-9). Similar trend was also found in the transfected A549 cells, that reported an activation of the cell proliferation, angiogenesis and blood coagulation (Figure 10; Table S8-9). However, no significant changes were found in cell proliferation, cell death rates, or cell cycle in any of the cells after miR-130a-3p transfection (Figure 11 A-C). Among the commonly deregulated genes, we found a significant up-regulation of the Rac Family Small GTPase 1 (RAC1) gene and neuropilin 2 (NRP2), which are involved in tumor migration and invasion in different tumors [38] including lung cancer [39,40]. In particular, Rac1 protein participates in the membrane protrusion formation, driving tumor cell migration [41] and the formation of membrane ruffles [42], also implicated in in the metastatic potential of tumor cells [43]. In addition, we also found deregulations of TME signaling involved in the onset and progression of cancer. Among these, angiogenesis was predicted in the miR-130-3p transfections. Currently, a number of studies support the role of miR-130a-3p in angiogenesis activation [44–46]. Among the commonly up-regulated genes after miR-130a-3p, we detected a significant increase of neuropilin 2 (NRP2) involved in the formation of new blood vessels [47]. In particular, very recently, Alghamdi and colleagues demonstrated as the NRP2 by RAC1 protein promote the adhesion and migration of the endothelial cells [48]. Additionally, we also observed an up-regulation of the Integrin Subunit Beta 1 (ITGB1), a protein involved in the vasculogenic mimicry, an alternative tumor mechanism of vessel-like structure formation [49].
All above data have been included in the final manuscript (Lines: 202-205; 209-212; 472-644; 683-697; 703-709; 782-787 and 792-803).
- Are the levels achieved by transfection of A549 cells with miRs comparable to what is seen in NSCLC patient samples?
Author response: We thank the reviewer for pointing out this relevant matter. As reported in the results section, we found significant exosomal enrichment of miR130a-3p and miR17-5p in the plasma of NSCLC patients who developed disease relapse after surgery. These results were also confirmed by ddPCR, at least for Exo-miR130-3p, reporting an increase of 2.6-fold. Thereafter, in order to understand their potential role in cancer progression, we transiently transfected the A549 and H2172 lung adenocarcinoma cell lines, with the specific mimics. The transfection efficacy was evaluated after 72 hours, reporting significant increases in miRs in all replicates of both cell lines (ranging from 460 to over 30,000, Figure 6). We are aware that NSCLC transfected cells showed higher miR enrichments than that found in patients, although this is not surprising as it compares an in vitro model with an in vivo one. Indeed, it should also be considered that in vitro forced expression of miRs on cells mimics only marginally the complexity of the in vivo system. Furthermore, in our experience, an approximately 3-fold miR increase would induce no significant effects on the cell lines. However, despite the limitations of our experimental design, the transfections caused significant gene modulations (Table S7), in turn inducing cellular changes in terms of apoptosis and cell cycle both at 72 and 96 hours (Figure 9 B,C; E-F) . Furthermore, our analysis allowed us to identify a set of genes targeted by miRs, that were shared by both two cell lines (Table 8).
- Authors discussed, a slight decrease in proliferation of CD3/CD28 stimulated PBMCs from healthy donors. However, the results shown in the supplementary figure do not look convincing enough to warrant that statement.
Author response: We thank the reviewer for this helpful indication and agree that the inhibitory effect was minimal and not significant. As correctly suggested, we repeated this test by increasing the cell-derived Exo-miR input (24 mL vs. the previous 12 mL of cell supernatant). However, the PBMC assessment showed no significant modulations in terms of inhibition/proliferation of stimulated or non-stimulated CD3/CD28 peripheral blood mononuclear cells (see attached word file), preventing us from understanding a potential oncogenic effect of Exo-miR-130a-3p. We hypothesize that this lack of effects may be due to the limitation linked to two-dimensional in vitro tests that occur outside a living system, and whose complexity do not simulate the real in vivo conditions. Indeed, we think that the Exo-miR effects might have a greater chance of success in co-culture of matched patient’s primary cell line and PBMC or 3D culture systems (i.e., organoids derived from patient’s biopsies). However, these very promising approaches are far from being carried out indeed, they would require specific approval by the Ethical Committee which unfortunately we do not have. Based on the above, we regret but we are unable to carry out such experiments. In conclusion, since our data do not show significant results and do not improve current knowledge, and also considering the limit of our experimental design (i.e., the two-Dimensional in vitro tests) we decided to remove it from the new version of the manuscript. However, we have also included this relevant point in the discussion as follows: “Last but not least, although numerous studies have already shown a connection between miR-130a-3p and angiogenesis, we have not performed in vitro studies proving its role in remodeling the TME.” (Lines: 749-751). Anyways, we thank the reviewer for this helpful advice, which will be considered for our new research project.
- It will be crucial to characterize changes in gene signatures not only in transfected A549 cells but also in PBMCs following co-culture with Exo-miR-130a-3p and miR17-5p. This experiment will certainly help in characterizing the immune-inhibitory role of miRs.
- One major drawback of the current study is that patients who received neoadjuvant chemotherapy are not selected. Is there any specific reason to exclude these patients? Normally, neo-adjuvant therapy is commonly the first line of treatment for NSCLC patients along with surgery.
Author response: We thank the reviewer for this useful suggestion. The decision to exclude patients who have undergone neoadjuvant treatment is closely linked to the objectives of the study. Indeed, any neoadjuvant treatment could hypothetically modify the behavior of the tumor cells and this might consequently change the Exo-miR enrichment and their effects. Furthermore, patients receiving neoadjuvant therapy are exclusively stage IIIA patients deemed operable and in our cohort those in this stage and whose plasma was available were a minority (8/110). For all these reasons we decided to exclude patients who had received neoadjuvant treatment which could be a confounding factor both at biological and statistical levels. However, as correctly suggested, we have detailed this relevant point in the “Materials and Methods” Section (‘Patient Selection and Collection of the Biological Specimen’ Paragraph) as follows: “iii) previous neoadjuvant therapy for NSCLC (due to its potential effect on the Exo-miR enrichment)”. (Lines: 129-130).

Reviewer 2 Report
The work presented is aimed to identify biomarkers predictive of relapse in early non-small cell lung cancer (NSCLC) patients. The authors utilize plasma and serum samples and combine two different approaches searching for differentially expressed micro-RNAs (miRNAs) and low molecular weight (LMW) proteins. The analysis of miRNAs contained in plasma exosomes is presented first and allowed the identification of 8 miRNAs up-regulated relapsed patients from a cohort of 67 patients and confirmed by ddCPR in 63 of them. Univariate Cox regression analysis considering clinical data identified one miRNA, miR-130a-3p whose expression significantly associated with worse prognosis. Multivariable regression analysis later identified that a combination of miR-130a-3p and miR-17-5p expression associated best with disease free survival, as also shown by Kaplan-Meier analysis. LMW protein expression in serum from 47 patients identified four peptides derived from Fibrinopeptide A significantly down-regulated in relapsed patients. Combination of the results obtained for miRNAs and peptides selected miR-130a-3p and one of the peptides (FpA 2-16) as a prognostic score to identify relapsed patients with significantly decreased disease-free survival. The possible role of the miRNAs identified was tested in vitro expressing them in the NSCLC A549 cell line. The consequence of over-expressing miR-130a-3p or miR-17-5p on the expression 770 genes involved in cancer progression was analyzed. The author found that miR-130a-3p expression lead to deregulation of angiogenesis, coagulation and metalloprotease pathways although did not alter cell cycle, apoptosis or cell death rates. The authors concluded that the score identified in blood samples may help clinicians in the prognosis of early-stage NSCLC patients.
The article presents an innovative strategy combining miRNA and peptide analyses as biomarkers of NSCLC prognosis. The study is very extensive and well presented. The data obtained are sound and support the conclusions of the authors. In addition, the results reported are of interest in NSCLC research and open new promise of clinical interest. There is just a minor point that could be improved in line 102 where “degradation proteins” might be “degradation of proteins”.
Other than that I recommend publication of the article in its present form.
Author Response
Reviewer #2
The article presents an innovative strategy combining miRNA and peptide analyses as biomarkers of NSCLC prognosis. The study is very extensive and well presented. The data obtained are sound and support the conclusions of the authors. In addition, the results reported are of interest in NSCLC research and open new promise of clinical interest. There is just a minor point that could be improved in line 102 where “degradation proteins” might be “degradation of proteins”. Other than that, I recommend publication of the article in its present form.
Dear reviewer#2 thank you for your suggestions; hereby our reply to the revision:
First of all, we thank the reviewer for expressing a favorable opinion on our study. We hope that our findings can be transferred to a clinical setting in the near future, helping the medical oncologist in the management of patients with non-metastatic NSCLC. We also acknowledge the reviewer for the correct observation on line 102, which we modified as suggested in “degradation of proteins” (Current line: 105).

Reviewer 3 Report
In the presented study, the Authors performed in-depth high-throughput analyses of plasma circulating markers, including exosomal microRNAs and peptidome aiming to identify potential prognostic biomarkers. The results and the approach of the study are very interesting and important, however, the study results are presented in a very confusing way. I think that the authors of this paper should go through the whole text in detail and make significant changes; otherwise, I think the manuscript is not acceptable for publication. Here are my comments:
- I would suggest changing of title from “ A Circulating Risk Score, Based on Combination of Exo-miR-130a-3p and Fibrinopeptide A, as Predictive Biomarker of Relapse in Early Stage Non-Small Cell Lung Cancer Patients” to “A Circulating Risk Score, Based on Combined expression of Exo-miR-130a-3p and Fibrinopeptide A, as Predictive Biomarker of Relapse in non-metastatic Non-Small Cell Lung Cancer Patients”, because authors claim that expression level of both is associated with higher risk of relapse and this should be clearly stated in the title. Also, early stage NSCLC in the title is discussable, because it is not clear what authors consider as early stage NSCLC. Tumors in Stage III are rather defined, large tumors with lymph nodes affected, and it is questionable if it could be considered as early stage. In the submitted manuscript, 12 % of patients included in the study were reported to be Stage III.
- In the NanoString Gene Expression Analysis paragraph (M&M), the authors should add how many biological replicates were included in the analysis. In the Results section, it is stated that four biological replicates were included, but that information is left out from M&M section. Also, the authors say that gene expression analysis of 770 genes involved in cancer progression processes was performed (line 172.), however in the Results section (line 392.) authors say “The expression patterns of 740 cancer progression genes… were studied”. These inconsistencies confuse readers so authors should decide and clearly state how many genes they studied in the end.
- In 2.7. paragraph, authors should specify threshold values chosen for the identification of signals significantly different in the two groups (lines 260-263), as well as details on how was the difference between group calculated.
- In the Figure 2. Authors could add calculated p-value. Also, font is rather small, so larger font size would be appreciated. Further, would recommend replacing y –graph label “# expressed miRs label” with “number of miRs expressed”
- In general, tables should appear after mentioned in the text, not before. For example, Table 2B. is on the page 8, and is first mentioned in the text on the page 9. The same replies to Table 2A. This table should be completely re-done because only first two columns respond to the results mentioned in the 3.2. paragraph Exo-miRNome profiling by microarray, where the table is mentioned and positioned. Other columns reported in this table should be made as single-standing tables in dedicated paragraphs. Also in this table, authors reported logHR for Exo-miRs detected by ddPCR and fpAs for 46 patients while in the section 3.4. and, as well as in Table S1, authors wrote that peptidome analysis was done on 47 patients, so why did authors excluded 1 patient from combined analysis in section 3.5?
- Table 2B. should be placed somewhere in 3.3. section. Authors should also report calculated values for other Exo-miRs in Table 2B. or remove them and leave only those that are significant. The way that this table is currently presented is not informative. Further, in my opinion, HR is more intuitive to understand than logHR (this applies to all tables), so authors could correct this. Also, in this table, authors first reported AIC values without any prior information and explanation (how it was calculated is missing in M&M section). Generally, in the entire manuscript explanation on how was circulating risk score calculated or with what method (eq. AIC) is missing. Since authors have decided to title the manuscript “ A Circulating Risk Score..” more focus should be added to the text regarding risk score calculation and significance of it.
- Figure 3. should be divided into two pictures. A) and B) part of Figure 3. should be left where they are, however, C) and D) should be placed after mentioned in text (paragraph 3.5, after Table 4. and new table from Table 2A, see comment 7.)). Bigger font sizes would be appreciated. Figure 4. has poor resolution and small font size. Text on signal peaks can not be seen.
- In lines 359-361 authors wrote: “In particular, these signals were significantly lower in relapsed patients (fold change 0.66, 0.31, 0.53 and 0.50, respectively) (Table 3; Figure 3.)”. I do not see anything in the Figure 3. that would match cited sentence?
- Table 3. lacks explanation of abbreviations used in table header. Readers might find it confusing that there is “significant” reduction in signals in relapsed patients when reported fold-change is a positive value, so, therefore, explanation of what Score stands for would be appreciated.
- In Table 4. Only calculated values should remain; not reported values (with - sign) should be removed.
Author Response
Reviewer #3
In the presented study, the Authors performed in-depth high-throughput analyses of plasma circulating markers, including exosomal microRNAs and peptidome aiming to identify potential prognostic biomarkers. The results and the approach of the study are very interesting and important, however, the study results are presented in a very confusing way. I think that the authors of this paper should go through the whole text in detail and make significant changes; otherwise, I think the manuscript is not acceptable for publication.
Dear reviewer#3 thank you for your comments; hereby our reply to the revision:
- I would suggest changing of title from “ A Circulating Risk Score, Based on Combination of Exo-miR-130a-3p and Fibrinopeptide A, as Predictive Biomarker of Relapse in Early Stage Non-Small Cell Lung Cancer Patients” to “A Circulating Risk Score, Based on Combined expressionof Exo-miR-130a-3p and Fibrinopeptide A, as Predictive Biomarker of Relapse in non-metastatic Non-Small Cell Lung Cancer Patients”, because authors claim that expression level of both is associated with higher risk of relapse and this should be clearly stated in the title. Also, early stage NSCLC in the title is discussable, because it is not clear what authors consider as early stage NSCLC. Tumors in Stage III are rather defined, large tumors with lymph nodes affected, and it is questionable if it could be considered as early stage. In the submitted manuscript, 12 % of patients included in the study were reported to be Stage III.
Author response: We thank the reviewer for this helpful suggestion. We agree to change the title of the paper, but at the same time we think that the definition of resectable NSCLC is more appropriate than “non-metastatic NSCLC”. Indeed, we have excluded cases of inoperable patients (i.e., Stage IIIB and Stage IV). The definition of non-metastatic patient also includes patients with locally advanced inoperable disease who were excluded from this work. In addition, we have also decided to exclude patients who had received neoadjuvant treatment (i.e., stage IIIA patients deemed operable) which were considered additional biological and statistical confounding factors. Indeed, any neoadjuvant treatment could hypothetically modify the behavior of the tumor cells and this might consequently change the Exo-miR enrichment and their effects. As correctly suggested and to clearly define the exclusion criteria of our resectable patient cohort, we have detailed these relevant points in the “Materials and Methods” Section (‘Patient Selection and Collection of the Biological Specimen’ Paragraph) as follows: “Exclusion criteria were: ….iii) previous neoadjuvant therapy for NSCLC (due to its potential effect on the Exo-miR enrichment); iv) death for a different cause other than NSCLC; v) unresectable IIIB and IV stage NSCLC (Tumor, Node, Metastasis (TNM) 7th International Union Against Cancer (UICC) Edition) [21]” (Lines: 129-132).
- In the NanoString Gene Expression Analysis paragraph (M&M), the authors should add how many biological replicates were included in the analysis. In the Results section, it is stated that four biological replicates were included, but that information is left out from M&M section. Also, the authors say that gene expression analysis of 770 genes involved in cancer progression processes was performed (line 172.), however in the Results section (line 392.) authors say “The expression patterns of 740 cancer progression genes… were studied”. These inconsistencies confuse readers so authors should decide and clearly state how many genes they studied in the end.
Author response: We apologize for our missing and confounding information. As correctly suggested we have included the number of biological replicates performed, and the text has been revised accordingly as follows: “For each analysis four biological replicates were performed.” (Line: 223).
Regarding the number of genes profiled by NanoString, it needs to consider that the nCounter PanCancer Progression Panel is a multiplex gene expression analysis that includes 740 cancer-related genes plus 30 internal reference controls used in the normalization process. These 740 genes are involved in each step of the cancer progression process including: angiogenesis, extracellular matrix remodeling (ECM), epithelial-to-mesenchymal transition (EMT) and metastasis. Therefore, in order to improve the readability, we have detailed this relevant information as follows: “Gene expression analysis was performed on transfected NSCLC A549 cell line using the nCounter platform (NanoString Technologies, Seattle, WA, USA) according to the manufacturer’s instructions. Specifically, we used the Human PanCancer Progression Panel that include 740 cancer-genes involved in the tumor progression processes such as angiogenesis, extracellular matrix remodeling, epithelial-to-mesenchymal transition, and metastasis, plus 30 internal reference controls.” (Lines: 184-189)
- In 2.7. paragraph, authors should specify threshold values chosen for the identification of signals significantly different in the two groups (lines 260-263), as well as details on how was the difference between group calculated.
Author response: We acknowledge the reviewer for these observations. The text has been revised accordingly as follows: “The output generated by Geena 2 [29] was used for pre-processing of spectra which were then analyzed by using the SAM (Significance Analysis of Microarrays) statistical method in order to identify sig-nals significantly different in the two groups (relapsed vs. non-relapsed patients). For each signal, the SAM analysis produced a q-value, described as a p-value adapted to a large number of comparisons, that was applied by using a threshold of 0.05 to assess the significance of the difference for that signal. The SAMR software, that implements the SAM method by using the R statistical language, was used to perform this task [30]. (Lines: 253-261)
- In the Figure 2. Authors could add calculated p-value. Also, font is rather small, so larger font size would be appreciated. Further, would recommend replacing y –graph label “# expressed miRs label” with “number of miRs expressed”
Author response: We acknowledge the reviewer for this correct observation. Therefore, we have modified the Figure 2 based on the suggestions.
- In general, tables should appear after mentioned in the text, not before. For example, Table 2B. is on the page 8, and is first mentioned in the text on the page 9. The same replies to Table 2A. This table should be completely re-done because only first two columns respond to the results mentioned in the 3.2. paragraph Exo-miRNome profiling by microarray, where the table is mentioned and positioned. Other columns reported in this table should be made as single-standing tables in dedicated paragraphs. Also in this table, authors reported logHR for Exo-miRs detected by ddPCR and fpAs for 46 patients while in the section 3.4. and, as well as in Table S1, authors wrote that peptidome analysis was done on 47 patients, so why did authors excluded 1 patient from combined analysis in section 3.5?
Author response: We acknowledge the reviewer for these observations. As correctly suggested, we modified the Table 2A separating into a single Table (i.e., Table 2, Table 3, Table 6) (Lines: 334-336; 346-348; 439-441) which have been reported alongside the cited text. We also apologize for our confounding information on number of patients analyzed. Indeed, one out of 47 patients, undergoing peptidome analysis, was excluded because the Exo-miR ddPCR value was not available. Therefore, the text has been revised accordingly as follows: “When peptides were combined with miRs for 46 of 47 patients (for one patient the ddPCR Exo-miR value was unavailable) …” (Lines: 442-443).
- Table 2B. should be placed somewhere in 3.3. section. Authors should also report calculated values for other Exo-miRs in Table 2B. or remove them and leave only those that are significant. The way that this table is currently presented is not informative. Further, in my opinion, HR is more intuitive to understand than logHR (this applies to all tables), so authors could correct this. Also, in this table, authors first reported AIC values without any prior information and explanation (how it was calculated is missing in M&M section). Generally, in the entire manuscript explanation on how was circulating risk score calculated or with what method (eq. AIC) is missing. Since authors have decided to title the manuscript “A Circulating Risk Score..” more focus should be added to the text regarding risk score calculation and significance of it.
Author response: We acknowledge the reviewer for these observations. As correctly suggested, we have modified the Table 2B by removing the non-informative Exo-miR data and including the data in the text. Furthermore, we agree with the reviewer on a more intuitive ‘Hazard Ratio’ value than the corresponding logHR one; therefore, we have replaced these values in all the Tables. Finally, in order to clear the building step of circulating risk score, we detailed it in the ‘Statistical analysis’ Paragraph (“Materials and Methods” Section), as follows: “Akaike’s Information Criterion (AIC), evaluated using the AIC function from the stat R package, was used to compare models, where lower AIC means better fitting of the model. The risk score to select patients according to their association to relapse was calculated by means of a weighted sum of the variables in the model, where the weights are the Cox regression coefficients (log-Hazard Ratio: HR). Predictors values are centered using their overall means.” (Lines: 269-274)
- Figure 3. should be divided into two pictures. A) and B) part of Figure 3. should be left where they are, however, C) and D) should be placed after mentioned in text (paragraph 3.5, after Table 4. and new table from Table 2A, see comment 7.)). Bigger font sizes would be appreciated. Figure 4. has poor resolution and small font size. Text on signal peaks can not be seen.
Author response: We acknowledge the reviewer for these observations. As correctly suggested, we have split Figure 3 into two single ones as follows: Figure 3 A-B (Lines: 372-385) and Figure 5 A-B (Lines: 457-470) and each of this has been reported alongside the cited text (Lines: 365 and 450). We also apologize for the poor resolution and small font size of the Figure 4; therefore, we have replaced it with a sharper image (Lines: 409-431).
- In lines 359-361 authors wrote: “In particular, these signals were significantly lower in relapsed patients (fold change 0.66, 0.31, 0.53 and 0.50, respectively) (Table 3; Figure 3.)”. I do not see anything in the Figure 3. that would match cited sentence?
Author response: We apologize for our confounding information. As corrected suggested, we removed the citation of Figure 3 that in the new manuscript version is Figure 4 (Lines: 409-431). Indeed, this figure reports an example of MS spectra from recurrent (A) and non-recurrent (B) NSCLC patients and it is correctly cited in lines 394-397
- Table 3. lacks explanation of abbreviations used in table header. Readers might find it confusing that there is “significant” reduction in signals in relapsed patients when reported fold-change is a positive value, so, therefore, explanation of what Score stands for would be appreciated.
Author response: We acknowledge the reviewer for these correct observations. The table and legend have been revised accordingly as follows: “Table 5. Results of the SAM analysis for the comparison between relapsed and non-relapsed patients. Only significant signals, having q-value < 5%, are reported. The SAM Score corresponds to a t statistics. Columns Num and Den reports the values of the numerator and denominator of the fraction calculated by the method for determining the Score. A negative Score, as well as a Fold change < 1, highlights a decrease of the values in the second group.” (Lines: 400-404)
- In Table 4. Only calculated values should remain; not reported values (with - sign) should be removed.
Author response: We acknowledge the reviewer for these correct observations. As suggested we removed any missing and not informative data, and the new table (Table 5) has been reported in the lines 404-405.

Round 2
Reviewer 1 Report
The authors have addressed all the comments. I believe this manuscript can be accepted in "Cancers".
Reviewer 3 Report
I have no further comments and suggest that the manuscript should be accepted in its present form.